# A Classification Bias and an Exclusion Bias Jointly Overinflated the Estimation of Publication Biases in Bilingualism Research

**DOI:** 10.3390/bs13100812

**Published:** 2023-10-01

**Authors:** Evelina Leivada

**Affiliations:** 1Departament de Filologia Catalana, Universitat Autònoma de Barcelona, 08193 Barcelona, Spain; evelina.leivada@uab.cat; 2Institució Catalana de Recerca i Estudis Avançats (ICREA), 08010 Barcelona, Spain

**Keywords:** bilingual advantages, bilingual disadvantages, publication bias, cognitive trade-off

## Abstract

A publication bias has been argued to affect the fate of results in bilingualism research. It was repeatedly suggested that studies presenting evidence for bilingual advantages are more likely to be published compared to studies that do not report results in favor of the bilingual advantage hypothesis. This work goes back to the original claim and re-examines both the dataset and the classification of the studies that were employed. We find that the exclusion of published works such as doctoral dissertations, book chapters, and conference proceedings from the original dataset significantly inflated the presumed publication bias. Moreover, the estimation of the publication bias was affected by a classification bias that uses a mega-category that consists of both null and negative outcomes. Yet finding evidence for a bilingual disadvantage is not synonymous with obtaining a result indistinguishable from zero. Consequently, grouping together null and negative findings in a mega-category has various ramifications, not only for the estimation of the presumed publication bias but also for the field’s ability to appreciate the insofar hidden correlations between bilingual advantages and disadvantages. Tracking biases that inflate scientific results is important, but it is not enough. The next step is recognizing the nested Matryoshka doll effect of bias-within-bias, and this entails raising awareness for one’s own bias blind spots in science.

## 1. Introduction

Publication biases are pervasive in all fields of science. It was found that empirical studies that observe a statistically significant effect are more likely to be published than studies that report no significant effect [1,2]. This publication bias in favor of positive findings has an impact on the distribution of published results, contributing to the canonization of dubious effects that find their way into the published record, while the results that challenge such effects may remain in the drawer [3]. As the absence of evidence is never evidence of absence, scholars who find null results are often required to go the extra mile, replicating across various conditions, and providing additional evidence for their failure to reject the null hypothesis, whereas significant evidence in favor of the same effect would require less defense [4].

The field of bilingualism is not an exception to the null result penalty. For at least a decade, various scholars have voiced concern that the literature on bilingualism and its benefits on various cognitive domains ([5,6]; see [7] for a recent overview of this relationship) is affected by a bias that favors the publication of positive results [8,9,10,11,12]. Succinctly put, the claim is that studies that find evidence for bilingual advantages are more likely to be published than studies that either find evidence for bilingual disadvantages or fail to find any effect. Research in this field was also affected by the file-drawer bias, according to which some scholars may submit for publication only the positive findings they obtain [9].

Putting things in perspective, this pattern does not reflect any field-specific bias: scientific discoveries usually start with positive results; negative findings typically emerge later, as the original effect is being scrutinized [13]. In this context, the phantom-like appearance of bilingual effects on cognition was explained, among other reasons, as the outcome of a time-induced balance between an early publication trend that favors positive results and the subsequent *Proteus phenomenon* [14], according to which initial positive results are rapidly followed by negative or null results, as the original phenomenon is being replicated across various conditions [15].

The landmark paper that is usually referenced when one mentions a publication bias in bilingual effects is de Bruin et al. (2015) [9]. This study quantified publication bias by tracing the fate of 104 conference abstracts. These abstracts were classified into four categories: (i) positive results (i.e., bilingual advantages), (ii) mixed results that partly support the bilingual advantage hypothesis (i.e., evidence for a bilingual advantage, albeit not in all tasks or populations), (iii) mixed results that partly challenge the bilingual advantage hypothesis (i.e., some inconsistent evidence for a bilingual advantage, but failure to find it in trials where it was expected), and (iv) negative or null results (i.e., evidence for a bilingual disadvantage or an effect indistinguishable from zero). Based on this classification system, it was found that conference abstracts fully supporting the bilingual advantage hypothesis were more than twice as likely to be published compared to those fully challenging this hypothesis [9].

Category (iv) groups together two outcomes: an effect that shows a bilingual disadvantage and an effect statistically indistinguishable from zero. This choice to group these two types of results together in a null/negative mega-category that is then juxtaposed with the ‘positive results’ category is not accidental. It has its roots in the field’s predominant view of bilingual effects: sometimes they are seen, but sometimes not [16]. This view has earned the overall body of evidence a reputation of inconsistent or contradictory results that are fueled by confirmation biases [17,18]: scholars who believe in the bilingual advantage hypothesis seem to find evidence in favor of it, while those who challenge it seem to frequently obtain results against it. Consequently, the gap in the interpretation of the overall body of evidence is so wide—even at the level of big systematic reviews and meta-analyses (cf. [19,20,21] vs. [11,22])—that it was recently argued that the available evidence tells two different tales, almost as if expectations were alchemically transmuted into results [18].

Across scientific fields, a good part of the established findings is byproducts of the prevailing biases [23]. Different types of biases come into play, including confirmation biases, publication biases, and file-drawer biases. This work brings to the fore a hitherto unanalyzed bias that boils down to how evidence is classified into different categories: a *classification bias*. This bias refers to how findings are assigned different labels, such as ‘evidence supporting the bilingual advantage hypothesis’ or ‘evidence challenging the bilingual advantage hypothesis’. To illustrate why the employed classification system is of utmost importance in the context of discussing bilingual effects on cognition, the *Jar Metaphor* [24] will be employed. 

Imagine two people: A and B. A and B are given an opaque jar with the instruction to determine what it has inside. The jar contains blue and red objects that look like pencils, but A and B are not told so. They decide to take up the challenge of finding what is in the jar and agree on the following discovery process: They will take turns retrieving objects from the jar, observing them carefully, and then providing a hypothesis about them. After retrieving two blue pencils, A proposes that the content of the jar is blue pencils. This is labeled the *blue pencil hypothesis*. B has found two red pencils, and they formulate a competing hypothesis called the *red pencil hypothesis*. At some point, B does not manage to take anything out of the jar (analogous to obtaining a null result), and based on this, they argue that since they have repeatedly failed to find a single blue pencil, the blue pencil hypothesis probably lacks the status of a robust phenomenon. Then, when it is again A’s turn to retrieve an object from the jar, they surprisingly find a red one. However, A suggests that this new finding does not challenge the validity of their blue pencil hypothesis because both A and B have previously agreed that there is a small margin of error in every pencil-selection event. Last, unexpectedly, B finds a bluish-looking pencil. However, after careful observation, they argue that this result cannot be taken as offering unequivocal support to the blue pencil hypothesis because the effect size of blueness is debatable. This pencil looks purple to B, and since purple contains red too, B interprets this finding as putative evidence in favor of their red pencil hypothesis. Eventually, it becomes evident that A and B engage with and interpret the same body of evidence in diametrically opposite ways.

The Jar Metaphor illustrates an important point: the theoretical light under which evidence is examined may color one’s perception of the obtained results [24]. This, in turn, affects the classification system one employs to make sense of what the results show in relation to the tested phenomenon. If one views bilingual advantages and disadvantages as boiling down to two competing, irreconcilable, or contradictory bodies of evidence, it is indeed reasonable that one organizes the results in terms of juxtaposing positive vs. null/negative categories, trying to *separately* assess their credibility. In contrast, if one employs the Jar Metaphor, there is no reason to call into question the validity of a subpart of the available evidence. Succinctly put, if the outcome of all or most pencil-selection events is true at the same time, regardless of which hypothesis one takes them to support, the source of contradiction does not lie in the results themselves. If the results are largely valid, what needs to be rethought is the “*either* blue *or* red” approach that both A and B have advanced. This is where an important lesson from biology that has repercussions for the classification system one puts forth enters the picture: observing an enhancement in one aspect of a system raises the question of what the compensation for this enhancement is, for one trait cannot increase without a decrease in another [25], given that organisms function as “integrated wholes” to use Darwin’s term [26]. 

Following this approach, it was recently suggested that bilingual advantages and disadvantages form *trade-offs* [24,27]: positive and negative effects are two aspects of the same coin (the latter understood here as an adaptation that responds to an environmental trigger; in this case, bilingualism). If we adopt this perspective, the practice of grouping null results together with negative results is not doing justice to the possible correlations between positive and negative outcomes. 

What does the picture of publication bias in bilingualism research look like if we alter the classification system? To address this question, the original results of de Bruin et al. (2015) were re-analyzed [9]. Leaving aside all other objections about possible reasons that may have impeded the publication of abstracts that do not find evidence in favor of bilingual advantages (e.g., there is no way of knowing how many abstracts that reported negative results were submitted for publication and rejected for independent reasons [28]), the aim is to re-quantify the available evidence, in an effort to determine whether the original claim about the fate of abstracts that do not report positive effects holds when one employs a different classification system that does not treat negative and null effects as parts of one homogeneous category that competes against positive effects.

## 2. Method

The dataset of de Bruin et al. (2015) was re-analyzed [9]. The abstracts were re-classified in terms of the evidence they adduced as follows: (i) positive results (i.e., bilingual advantages), (ii) mixed results, (iii) negative results (i.e., bilingual disadvantages), and (iv) results that point to group differences that were indistinguishable from zero (i.e., null results). While the original classification of de Bruin et al. (2015) split the category of mixed results into two different categories (i.e., results partly supporting the bilingual advantage hypothesis vs. results partly challenging it), the authors describe both categories as ultimately involving mixed results [9]. Since evaluative statements such as ‘partly supporting’ and ‘partly challenging’ are largely a matter of interpretation given to the same outcome (i.e., some, but not all, results of a given study support the bilingual advantage hypothesis), the two mixed categories are grouped together in the analyses that follow. This decision is based on two reasons: first, de Bruin et al. do not offer any objective criteria for drawing a distinction between the two categories; and second, most of these abstracts are no longer available, and this hinders the ability to re-evaluate their contents as predominantly supporting vs. challenging (a matter that is further analyzed in Discussion). The updated distribution of conference abstracts in terms of results, together with their status as published or not is given in Table 1. 

To perform the analyses, a Bayesian approach was used. Critics of the bilingual advantage hypothesis have voiced the opinion that a Bayesian approach is more appropriate than frequentist approaches when one seeks to examine the effects of bilingualism on cognition because it enables us to estimate how likely the data are to occur under the null vs. the alternative hypothesis [18]. More specifically, the Bayes Factor (BF) calculates the conditional probability of the occurrence of the analyzed data given the null vs. the alternative hypothesis [29]. Following standard thresholds, a BF_10_ factor of 1 suggests that the analyzed data are almost equally likely under either the null or the alternative hypothesis, providing inconclusive evidence. BFs larger than 3 are generally taken as weak-to-moderate evidence in favor of the alternative, values larger than 10 as strong evidence, and values larger than 30 as very strong evidence, while a BF_10_ > 100 can be interpreted as extreme support in favor of the alternative hypothesis [30]. The inverse of these cut-offs (<1/3, 1/10, 1/30, and 1/100, respectively) is interpreted as providing different degrees of support for the null hypothesis. As the interpretation of BFs occurs on a continuous scale, values between 0.33 and 3 are considered inconclusive, not offering robust evidence for either hypothesis [31].

Since the aim is to determine whether one category of findings fares better than the others in terms of publication status, four separate analyses (one per category) were run, plus an overall analysis that involves all categories together. The null hypothesis is that there are no statistically significant differences in terms of publication status in any category. To confirm the claim of de Bruin et al. about a publication bias that favors positive results, evidence for the alternative hypothesis should be found when analyzing category (i) that concerns bilingual advantages: The proportion of published studies should be significantly higher in this category.

For the purposes of the present analyses, the variable ‘effects’ was coded in terms of the following values: positive, negative, mixed, or null (Figure 1). The variable ‘published status’ was coded as 1 (published) or −1 (not published). The analyses were run using jamovi, version 2.2 [32]. Figure 1 shows the distribution of the published vs. unpublished abstracts following the new classification system.

## 3. Results

The first analysis concerns the overall effects: Merging the four categories into one, do the results provide evidence for significant differences in terms of publication status? A Bayesian one-sample *t*-test suggests that the answer is negative. A BF_10_ = 0.109 suggests that the data are more probable under the null hypothesis (Figure 2). In this case, jamovi labels the evidence in favor of the null hypothesis as moderate (panel ‘Sequential Analysis’). In the panel ‘Prior and Posterior’, the priors show the allocation of credibility across the continuum of possible parameter values before the analyzed data are introduced [33]. The posterior represents the re-allocated probability after the data are analyzed. The panel ‘Bayes Factor Robustness Check’ shows how the interpretation of the BF changes if different priors are chosen. In this case, this interpretation ranges from providing inconclusive to moderate to strong evidence but always in favor of the null. 

Observing that the overall dataset does not show evidence for significant differences in terms of the portion of studies that were vs. were not published does not directly speak to the main claim regarding publication bias because the latter seems category specific (i.e., it depends on the type of findings reported in the abstracts). Therefore, all four categories were separately analyzed. Starting from the critical category of positive findings, where a strong effect of publication bias was expected, a Bayesian one sample *t*-test suggests that the evidence that the data are more probable under the alternative hypothesis is anecdotal or inconclusive: BF_10_ = 1.891. As Figure 3 shows, this evidence is classified as anecdotal regardless of the different possible priors that one uses. 

The picture remains the same in terms of not providing strong evidence for a publication bias when one analyzes the category that involves mixed results. A Bayesian one sample *t*-test suggests that the data seem more probable under the null hypothesis, BF_10_ = 0.258. Figure 4 suggests that, unlike the previous analysis of positive results, where the evidence was inconclusive, for the category of mixed results, we find moderate evidence in favor of null. 

The two most critical categories concern null and negative effects, respectively. Recall that these findings were grouped together in de Bruin et al. [9], but they are classified and analyzed as separate categories in the present work. The expectation is that the analyses of these categories will provide strong evidence for the alternative hypothesis since previous estimates suggested that only around 29% of the abstracts falling in these categories were published. Figure 5 and Figure 6, for null and negative results, respectively, suggest that this prediction is not borne out. In the case of null results (Figure 5), the obtained BF_10_ = 1.693 corresponds to anecdotal evidence in favor of the alternative, which remains stable across different priors. In the case of negative results (Figure 6), which is the least plentiful category of all, BF_10_ = 0.428 amounts to anecdotal evidence in favor of the null hypothesis.

A Bayesian ANOVA that examines the effect of ‘category’ on publication status reveals only weak evidence in favor of the alternative hypothesis that there is such an effect (BF_10_ = 3.65). Importantly, the post hoc comparisons given in Table 2 show that the robust difference is not found between positive vs. negative results (BF_10_ = 0.513) but between positive vs. null results (BF_10_ = 9.297). The differences between categories are shown in Figure 7.

As mentioned in the Introduction, a good part of the established scientific findings is byproducts of the prevailing biases, and this holds across scientific fields [23]. What is often unacknowledged is that prevailing biases regularly come in the form of a nested Matryoshka doll effect (i.e., *bias-within-bias*), such that their synergistic interplay may give rise to a false or overinflated effect that finds its way into the published record. To provide a concrete example, the analyses presented above altered the classification system originally employed by de Bruin et al. but did not alter the original dataset. Focusing on the dataset itself, it seems that a second bias comes into play: the exclusion of published works such as doctoral dissertations, book chapters, and conference proceedings from the original dataset has significantly inflated de Bruin et al.’s estimation of the presumed publication bias. 

Table 3 and Table 4 illustrate this issue. While all the conference abstracts listed below are classified by de Bruin et al. as not published/accepted for publication on or before 20 February 2014, the Reference column in Table 3 and Table 4 reveals that a thorough literature review carried out in 2023 provides evidence that most of these abstracts were already published by 2014, in the form of dissertations, book chapters, or journal articles. While a number of these publications indeed do not fall within the cut-off point of de Bruin et al., most of them do.

Taking Table 1, Table 3 and Table 4 together, 33/40 abstracts in the category ‘positive results’ were published by 2023, while for the category ‘negative and null results’, the published abstracts are 14/17, or 82.5% and 82.35% for each category respectively. Ιn other words, even if we maintain the original classification and the null/negative mega-category, the updated numbers do not replicate the result of de Bruin et al. [9] regarding a significant effect of ‘category’ on publication status. As Figure 8 shows, using the updated dataset, a Bayesian independent samples *t*-test finds weak-to-moderate evidence in favor of the null hypothesis that there is no difference between the two critical categories, ‘positive’ vs. ‘null/negative’ in terms of publication status (BF_10_ = 0.288). Importantly, this higher probability of the data under the null cannot be attributed to those studies that were published in or after 2014, that is, studies that fall outside de Bruin et al.’s cut-off point. Even if these studies are removed from the picture, taking only the studies published up to 2013, which could have been included in the original dataset but were not, the presumed effect against null/negative results is still not found. Again, a Bayesian independent samples *t*-test finds weak-to-moderate evidence in favor of the claim that the data are more probable under the null (BF_10_ = 0.306, Figure 9). To put this last result in comparison with the frequentist analysis that de Bruin et al. ran, a binomial logistic regression confirms that there is no significant difference between the two categories, Z = 0.393, *p* = 0.694.

## 4. Discussion

It was repeatedly argued that the field of bilingualism favors the publication of studies that find evidence for bilingual advantages. To provide concrete numbers, it was reported that only an estimated 29% of a set of conference abstracts fully challenging the bilingual advantage hypothesis were eventually published, compared to 68% of abstracts supporting bilingual advantages [9]. Undoubtedly, these marked differences are important, and taking stock of publication biases is a critical step toward scientific progress. In this context, it is unsurprising that the results of the landmark paper that quantified this publication bias (i.e., de Bruin et al. 2015) have been widely cited (i.e., 636 times according to Google Scholar, with a total of 11206 views and downloads according to the journal’s metrics) as providing robust evidence for a publication bias in bilingualism research that disfavors null and negative results. 

As the previous section suggested, upon changing the classification system in order to avoid equating null effects and bilingual disadvantages, the results paint a picture that is different from the one reported by de Bruin et al. [9]. This new picture largely corresponds to the typical null result penalty—which is not field-specific—without, however, providing solid evidence for a bias that disfavors the publication of negative results. It seems that the estimation of the presumed publication bias that disfavors negative results was affected by a classification bias that uses a mega-category that consists of *both* null and negative outcomes. The claim is that a publication bias affected the publication of studies that “fully challenge” ([9], p. 3) the bilingual advantage hypothesis, but Table 2 shows that the significant difference boils down to positive vs. *null* results, and null results cannot challenge or disprove the alternative hypothesis. Absence of evidence is not evidence of absence. In frequentist statistics, “null results are formally meaningless: there was insufficient evidence in the dataset to reject the null hypothesis, but *that is all one can say*” ([51], p. 2; emphasis added).

Leaving the formal interpretation of null results aside, why exactly is a classification system that juxtaposes positive vs. null/negative results biased? As argued in the Introduction, if one views bilingual advantages and disadvantages as two competing, irreconcilable, or contradictory bodies of evidence, it is indeed reasonable that one organizes the results in such a way, suggesting that the phenomenon described by a subpart of the available evidence either does not exist at all or occurs in a phantom-like way, in restricted and insofar undetermined circumstances [17]. However, another conceptualization of the available evidence is possible. It was recently argued that this practice of differently engaging with subparts of the available evidence amounts to the fallacy of Stacking the Deck, which significantly impacts the field’s ability to synthesize and interpret the results into an overarching, ecologically valid theory [24]. Continuing with the Jar Metaphor presented in the Introduction, A and B have stacked the deck because they focus on different portions of the available evidence. While A focuses on those results that support their blue pencil hypothesis, B grants more prominence to the results that favor their red pencil hypothesis. However, a different approach is possible. Imagine that the objects taken out of the jar are not separate pencils but parts of a single chunk of wood that ranges from blue to red: behind the superficial impression of separate blue and red objects, there is a continuum of effects that boil down to the same cause [24]. 

Put another way, if one accepts that positive and negative outcomes form *trade-offs* as a result of the same environmental trigger (i.e., bilingualism), it becomes clear that pitting blue pencils (say, positive findings) against red pencils (say, negative findings), and then differently assessing the credibility of these two bodies of evidence as if they were unconnected amounts to a biased classification system that breaks the available evidence into fragments. In contrast, the trade-off approach seeks to replace this fragmented view of bilingual effects as stand-alone outcomes that are scattered across cognitive domains with the view that positive and negative outcomes may form correlations. The latter is so far undetermined, largely due to the fact that an overarching theory of bilingual effects is still missing [52], as a consequence of the fact that the available evidence tells two different stories depending on who is narrating (cf. [18] for a similar claim). This new perspective has the potential to revolutionize the way the field conceptualizes bilingual advantages and disadvantages, paving the way for a new era that will leave such labels behind. Importantly, this view agrees with the claim of de Bruin et al. that publication biases may be worsened by researchers’ prejudices and agendas [9]. The take-home message is that various biases come into play during the research process, and these frequently cause overestimation of some effects [53]. The classification bias that gave rise to the presumed publication bias seems to be one such example. 

In relation to the need to rethink de Bruin et al.’s classification, an anonymous reviewer suggests that there are so few negative findings according to the original dataset, and they split 50/50 in terms of publication status (Table 1), so they cannot materially affect the outcome. They also suggest that it is presumably because there are so few negative results that de Bruin et al. combined them with null results to begin with. Three separate issues merit clarification here. First, the soundness of the classification system is not evaluated against the outcome but against the claim put forth on the basis of the outcome. Assuming that the category of negative results was indeed split in terms of publication status, what substantiates de Bruin et al.’s main claim that a publication bias has affected the publication of studies that “fully challenge” ([9], p. 3) the bilingual advantage hypothesis if the ‘fully challenging’ category is perfectly balanced for publication status? Second, the assumption mentioned in the previous point is unwarranted. Abstracts that report bilingual disadvantages are not really split 50/50 in terms of publication status: While de Bruin et al. do not specify which four abstracts within the null-negative mega-category they count as non-null (i.e., as finding bilingual disadvantages), once we take Table 3 into account, the previously perfectly split sample is no longer perfectly split. Two abstracts that make explicit reference to bilingual disadvantages (i.e., ‘Bilingualism and the acquisition of number skills’ and ‘A bilingual disadvantage in linguistic perspective adjustment’) were classified in the original dataset as not published, but they were published. The former was published in 2010, so it falls within de Bruin et al.’s February 2014 cut-off point, while the latter was published in May 2014, 6 months before de Bruin et al. Third, and more importantly, the fact that only four abstracts were included in the ‘bilingual disadvantage’ sub-category to begin with—eventually giving rise to an extremely small sample that presumably had to be combined with the null results, as the reviewer suggests—does not mean that there were no other good candidates for this category. Re-evaluating the content of all the screened abstracts is difficult because these conferences took place 10–20 years ago, and most abstract links no longer work. However, based on the very few links that still work in August 2023, it seems that several abstracts that were presented in the conferences de Bruin et al. analyzed found bilingual disadvantages but were not included in the dataset. One example is ‘Noun-phrase production in bilinguals’ ([54], published as [55]), which found a bilingual disadvantage in response onset times in picture-naming tasks. While de Bruin et al. did not count the “effects of bilingualism in lexical tasks without a clear executive-control component” ([9], p. 2) as relevant, it is not at all uncontroversial that there is such a thing as a *clear* executive-control component. Much research has suggested that executive functions contribute to successful word retrieval [56,57,58], so naming tasks could, in principle, be included. In sum, it seems that the sample size was critically reduced by excluding both a) abstracts that find certain bilingual disadvantages possibly related to executive control and b) certain types of published outputs that derive from the included abstracts. In this context, merging results of different ilk in one null-negative mega-category does not remedy the self-inflicted small sample size problem; it simply adds a confound to the analysis, not allowing us to estimate the true size of the presumed publication bias against negative results. 

Taking stock of the second bias, it seems that by virtue of excluding certain published outputs from the original dataset, the presumed publication bias was inflated, giving rise to an effect that does not seem to exist once all the relevant published outputs are factored in. Again, it is necessary to highlight that absence of evidence is not evidence of absence. It could be the case that a publication bias against results challenging the bilingual advantage hypothesis truly exists. Yet, this claim must be supported by evidence, and the dataset of de Bruin et al. does not offer this evidence once properly updated. 

Crucially, no scientific reason is offered for the inclusion of only journal articles into the original dataset [9]. The practice of granting a more prominent position to journal articles to the detriment of the overall visibility of other types of publications is not untypical in many fields of science and amounts to an *exclusion bias* (also known as *location bias*). This is a type of publication bias that targets studies found in the so-called grey literature (i.e., conference proceedings, dissertations, technical reports, book chapters, etc.). These studies are often excluded from meta-analyses, quantitative analyses, and systematic reviews with grievous consequences. For instance, excluding grey literature, and restricting resources to a few widely used platforms may lead to biased estimates of effect sizes [59]. Moreover, in the field of health care, it was found that by excluding grey literature, meta-analyses are likely to artificially inflate the benefits of a health intervention [60]. Evidently, excluding a portion of the published results without adequate justification is an instance of publication bias that may influence the magnitude and credibility of an alleged effect.

Having established that various biases are behind the overinflation of the presumed publication bias against results that challenge the bilingual advantage hypothesis, it is useful to take a step back and ponder whether there is any evidence left to substantiate the original claim. For instance, the meta-analysis of de Bruin et al. (2015) provides a funnel plot that shows a clear asymmetry [9]. Succinctly put, the claim is that if there was no publication bias, the plot should be symmetric because studies with larger standard errors (i.e., found at the low part of the y-axis) should spread symmetrically at the bottom of the graph. If the dots are found only in the bottom right, studies with large standard errors and small effect sizes are missing, hence the asymmetry.

Several issues merit unpacking with respect to what kind of evidence the funnel plot asymmetry provides. First, it does not explain what *category* of results the missing dots correspond to. The previous section has indeed established that the field of bilingualism research is not an exception to the null result penalty. However, observing that null results may be harder to publish does not validate the claim that the field is biased against results that find evidence for bilingual *dis*advantages. 

Second, the expected symmetry is based on the critical assumption that there is no relationship between effect size and sample size. However, as Simonsohn (2017) argues, this assumption is false if researchers use larger samples to investigate effects that are harder to detect or when they simply adjust sample size based on what they know from previous literature [61]. Discussing the funnel plot of de Bruin et al., Simonsohn notes that since some studies tap into the performance of healthy young bilingual adults in the Simon task, while others examine whether bilingualism predicts age of diagnosis in patients with Alzheimer’s disease, the funnel plot would be “diagnostic of publication bias only if the sample sizes researchers used to study these disparate outcomes are in no way correlated with effect size” [61]. This seems to be an unwarranted assumption.

Third, a funnel plot asymmetry can arise for several reasons that may have nothing to do with publication bias. While a publication bias may give rise to an asymmetry, an asymmetry is not always due to a publication bias; it can emerge due to other reasons [19]. In Simonsohn’s (2017) simulation of 100 studies, each with a true effect size drawn from d~N (.6,.15), all results were reported, but many asymmetrical funnel plots were generated [61]. This suggests that the funnel plot asymmetry does not provide evidence for the claim that the field of bilingualism research is biased against publishing negative results that document bilingual disadvantages.

To sum up, de Bruin et al.’s (2015) concluding remark raises a valid, important point, which is confirmed by the results of the present work: “All data, not just selected data that supports a particular theory, should be shared, and this is especially true when it comes to data regarding issues that have enormous societal relevance and implications, such as bilingualism” ([9], p. 8). It is true that most scientists have blind spots, prior beliefs, and theories that affect the way they engage with scientific findings. In this context, raising awareness for one’s own bias blind spots is important for scientific progress.

## Figures and Tables

**Figure 1 behavsci-13-00812-f001:**
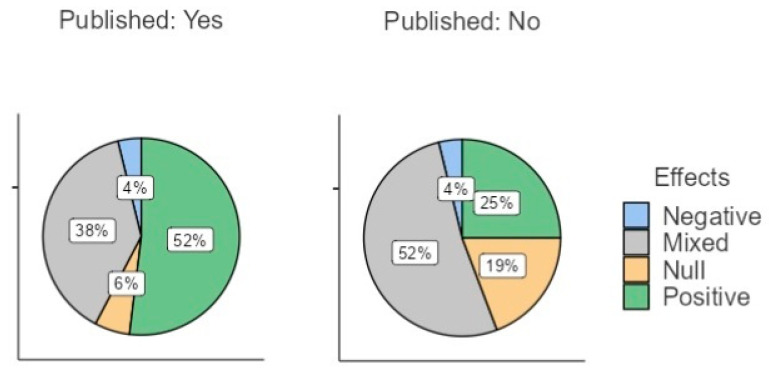
Distribution of abstracts in terms of findings, split for published vs. unpublished status.

**Figure 2 behavsci-13-00812-f002:**
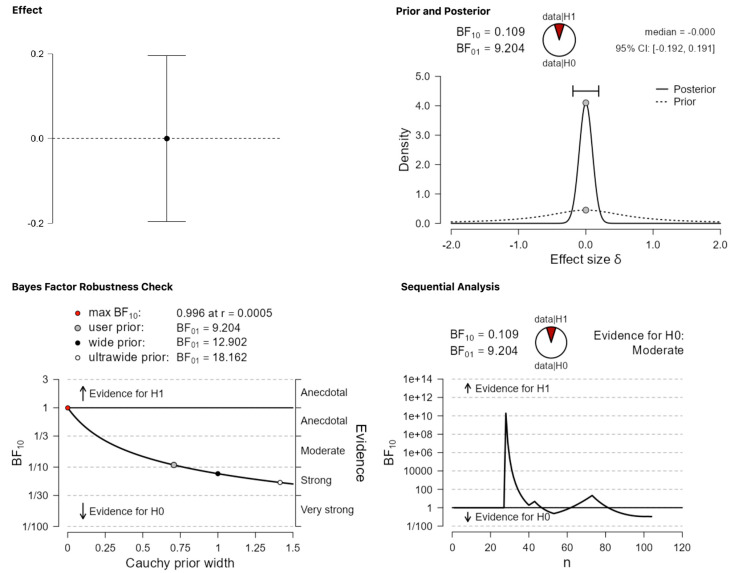
Publication differences in the overall pool of data (all categories together).

**Figure 3 behavsci-13-00812-f003:**
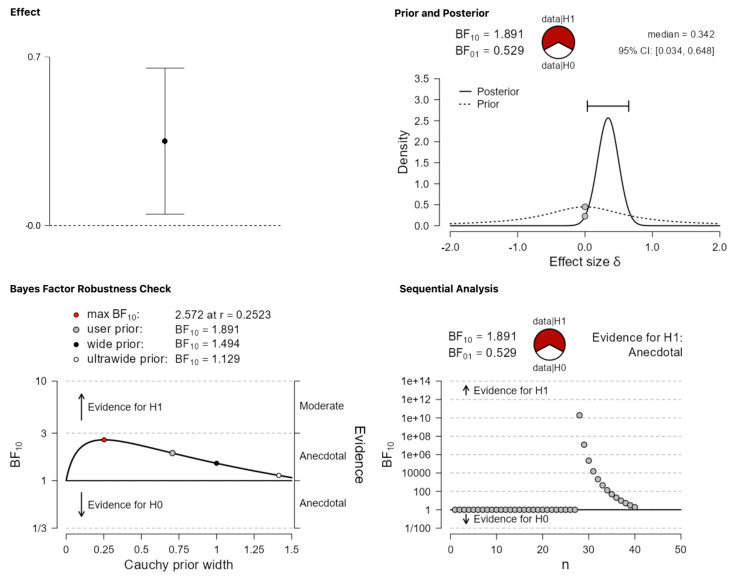
Publication differences in category (i) ‘positive results’.

**Figure 4 behavsci-13-00812-f004:**
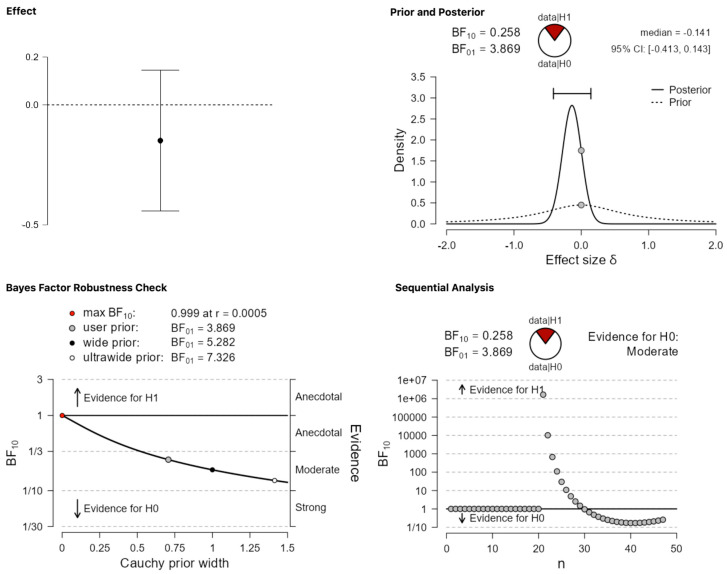
Publication differences in category (ii) ‘mixed results’.

**Figure 5 behavsci-13-00812-f005:**
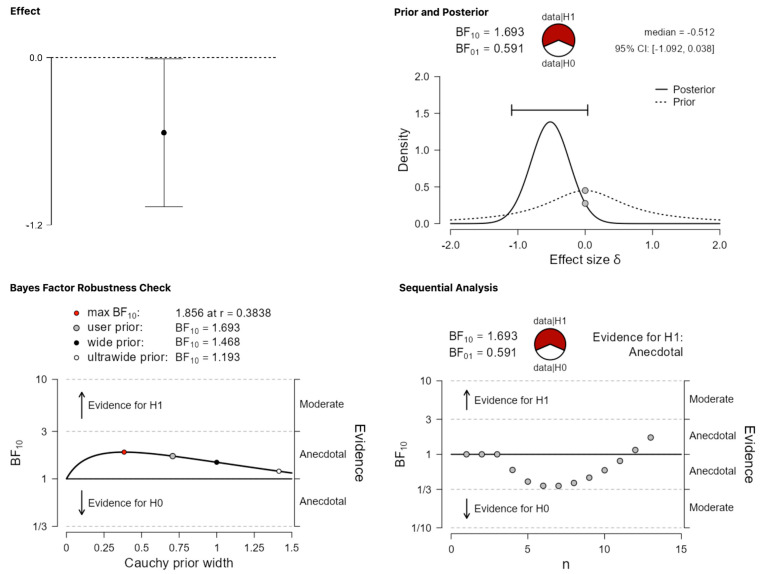
Publication differences in category (iii) ‘null results’.

**Figure 6 behavsci-13-00812-f006:**
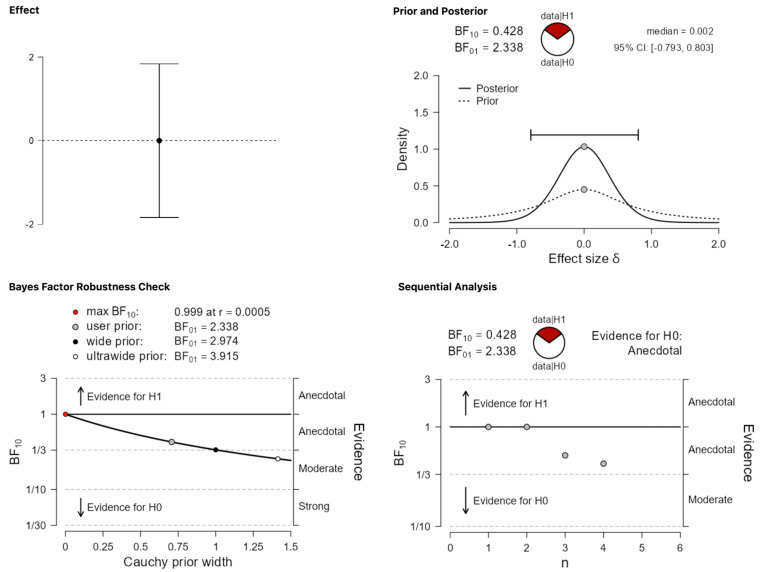
Publication differences in category (iv) ‘negative results’.

**Figure 7 behavsci-13-00812-f007:**
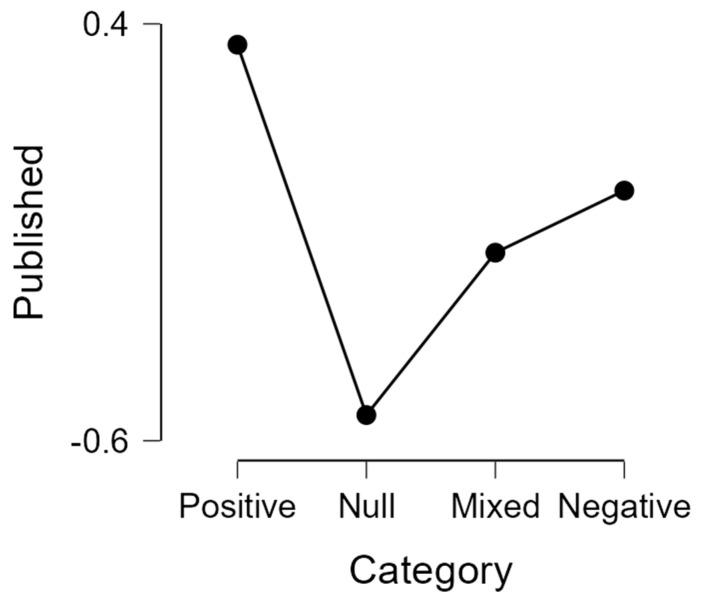
The effect of ‘category’ on published status.

**Figure 8 behavsci-13-00812-f008:**
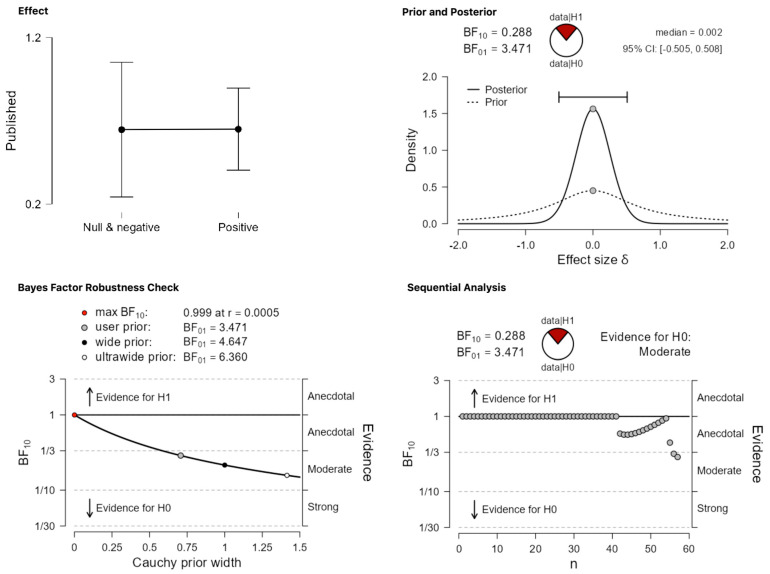
The effect of ‘category’ in the updated dataset when maintaining the original classification system. The comparison is between the critical categories ‘positive’ vs. ‘null/negative’.

**Figure 9 behavsci-13-00812-f009:**
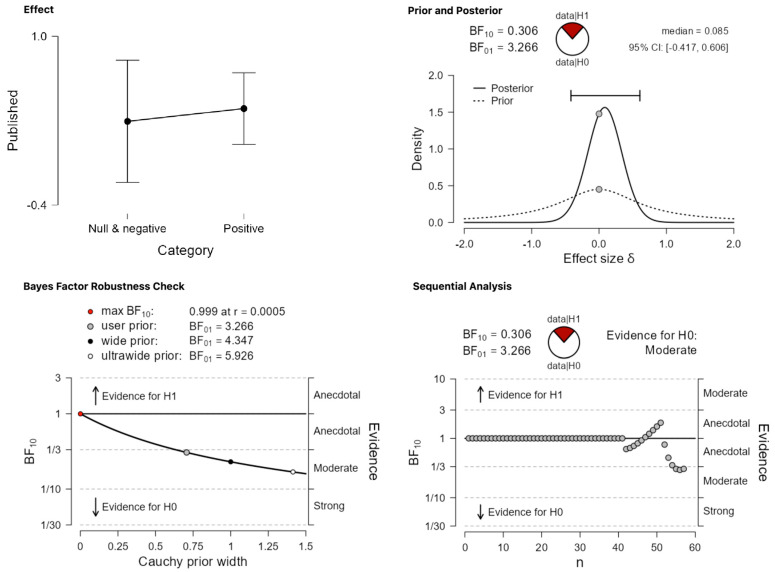
The effect of ‘category’ in the updated dataset when maintaining the original classification system. The comparison is between the critical categories ‘positive’ vs. ‘null/negative’. This modified version of the updated dataset includes only the studies published within the cut-off point (by February 2014).

**Table 1 behavsci-13-00812-t001:** The dataset (based on the numbers reported in de Bruin et al. 2015) [9].

Positive	Mixed	Negative	Null	Published
40				27/40
	47			20/47
		4		2/4
			13	3/13
**TOTAL**				52/104

**Table 2 behavsci-13-00812-t002:** Post hoc comparisons for the effect of ‘category’ on publication status. The posterior odds were corrected for multiple testing by fixing to 0.5 the prior probability that the null hypothesis holds across comparisons [34]. Individual comparisons are based on the default *t*-test with a Cauchy (0, r = 1/sqrt(2)) prior. The “U” in the BF denotes that it is uncorrected.

Post Hoc Comparisons—Category					
		Prior Odds	Posterior Odds	BF_10,U_	Error%
Positive	Null	0.414	3.851	**9.297**	7.08 × 10^−6^
	Mixed	0.414	1.057	2.551	9.37 × 10^−4^
	Negative	0.414	0.213	0.513	5.39 × 10^−4^
Null	Mixed	0.414	0.241	0.583	0.0109
	Negative	0.414	0.263	0.635	3.30 × 10^−4^
Mixed	Negative	0.414	0.186	0.450	2.30 × 10^−5^

The most robust difference appears in bold.

**Table 3 behavsci-13-00812-t003:** The list of unpublished abstracts that do not find any evidence for bilingual advantages [9] and their updated publication status in 2023.

Unpublished Abstracts That Do Not Find Evidence for Bilingual Advantages (List Taken from [9])	Status in 2023	Reference
Guagnano, D., Rusconi, E., Job, R., & Cubelli, R. (2009). Bilingualism and the acquisition of number skills.	Published	Guagnano, D. (2010). *Bilingualism and cognitive development: a study on the acquisition of number skills.* PhD thesis, University of Trento. [35]
Humphrey, A. D., & Valian, V. V. (2012). Multilingualism and cognitive control: Simon and Flanker task performance in monolingual and multilingual young adults.	Published	Valian, V. (2015). Bilingualism and cognition. *Bilingualism: Language and Cognition* 18, 3–24. [36]
Inurritegui, S., & D’Ydewalle, G. (2008). Bilingual advantage inhibited? Factors affecting the relation between bilingualism and executive control.	Published	Inurritegui, S. (2009). Bilingualism and cognitive control. PhD thesis, KU Leuven. [37]
Kennedy, I. (2012). Immersion education in Ireland: Linguistic and cognitive skills.	Published	Kennedy, I. (2012). Irish medium education: cognitive skills, linguistic skills, and attitudes towards Irish. PhD thesis, Bangor University. [38]
Mallery, S. T. (2005). Bilingualism and the Simon task: Congruency switching influences response latency differentially.	Not published	
Mallery, S. T., Llamas, V. C., & Alvarez, A. R. (2006). Performance Advantage on the Tower of London-DX for Monolingual vs. Bilingual Young Adults.	Not published	
Perriard, B., & Camos, V. (2011). Working memory capacity in French-German bilinguals.	Published	Perriard, B. (2015). L’effet du bilinguisme sur la mémoire de travail: comparaisons avec des monolingues et étude du changement de langue dans des tâches d’empan complexe. PhD thesis: Université de Fribourg. [39]
Ryskin, R. A., & Brown-Schmidt, S. (2012). A bilingual disadvantage in linguistic perspective adjustment.	Published	Ryskin, R. A., Brown-Schmid, S., Canseco-Gonzalez, E., Yiu, L. K. & Nguyen, E. T. (2014). Visuospatial perspective-taking in conversation and the role of bilingual experience. *Journal of Memory and Language* 74, 46–76. [40]
Sampath, K. K. (2003). Effects of bilingualism on intelligence.	Published	Sampath, K. K. (2005). Effect of bilingualism on intelligence. In J. Cohen, K. T. McAlister, K. Rolstad & J. MacSwan (eds.), *Proceedings of the 4th International Symposium on Bilingualism*, 2048-2056. Somerville, MA: Cascadilla Press. [41]
Tare, M., & Linck, J. A. (2011). Bilingual cognitive advantages reduced when controlling for background variables.	Not published	
Vongsackda, M., & Ivie, J. L. (2010). Working memory differences between monolinguals and bilinguals.	Published	Vongsackda, M. (2011). Working memory differences between monolinguals and bilinguals. MA thesis: California State University. [42]
Weber, R. C., Johnson, A., & Wiley, C. (2012). Hot and cool executive functioning advantages in bilingual children.	Published	1. Weber R. C. (2011). How hot or cool is it to speak two languages: Executive function advantages in bilingual children. PhD thesis: Texas A&M University. [43] 2. Weber, R. C., Johnson, A., Riccio, C. A. & Liew, J. (2016). Balanced bilingualism and executive functioning in children. *Bilingualism: Language and Cognition* 19, 425–431. [44]

**Table 4 behavsci-13-00812-t004:** The list of unpublished abstracts that fully support bilingual advantages [9] and their updated publication status in 2023.

Unpublished Abstracts That Find Evidence for Bilingual Advantages (List Taken from [9])	Status in 2023	Reference
Bak, T. H., Everington, S., Garvin, S. J., & Sorace, A. (2008). Differences in performance on auditory attention tasks between bilinguals and monolinguals.	Published	Bak, T. H.,Vega-Mendoza, M. & Sorace, A. (2014). Never too late? An advantage on tests of auditory attention extends to late bilinguals. *Frontiers in Psychology* 5: 485. [45]
Barac, R., Moreno, S., & Bialystok, E. (2010). Inhibition of responses in young monolingual and bilingual children: Evidence from ERP.	Published	Barac, R., Moreno, S., & Bialystok, E. (2016). Behavioral and electrophysiological differences in executive control between monolingual and bilingual children. *Child Development* 87(4), 1277–1290. [46]
Boros, M. Marzecova, A., & Wodniecka, Z. (2011). Investigating the bilingual advantage on executive control with the verbal and numerical Stroop task: Interference or facilitation account?	Not published	
Chin, S., & Sims, V. K. (2006). Working memory span in bilinguals and second language learners.	Not published	*
Díaz, U., Facal, D., González, M., Buiza, C., Morales, B., Sobrino, C., Urdaneta, E., & Yanguas, J. (2011). The use of bilingualism and occupational complexity measures as proxies for cognitive reserve: results from a community-dwelling elderly population in the north of Spain.	Not published	
Duncan, H., McHenry, C., Segalowitz, N., & Phillips, N. A. (2011). Bilingualism, aging, and language-specific attention control.	Not published	
Friesen, D. C., Hawrylewicz, K., & Bialystok, E. (2012). Investigating the bilingual advantage in a verbal conflict task.	Not published	
Grote, K. S., & Chouinard, M. M. (2010). The potential benefits of speaking more than one language on non-linguistic cognitive development.	Published	Grote, K. S. (2014). The cognitive advantages of bilingualism: A focus on visual-spatial memory and executive functioning. PhD thesis: University of California, Merced. [47]
Luo, L., Seton, B., Bialystok, E., & Craik, F. I. M. (2008). The role of bilingualism in retrieval control: Specificity and selectivity.	Published	Luo, L., Craik, F. I. M., Moreno, S., & Bialystok, E. (2013). Bilingualism interacts with domain in a working memory task: Evidence from aging. *Psychology and Aging 28*, 28–34. [48]
Luo, L., Sullivan, M., Latman, V., & Bialystok, E. (2011). Verbal recognition memory in bilinguals: The word frequency effect.	Not published	
Sullivan, M., Moreno, S., & Bialystok, E. (2010). Effects of early-stage L2 learning on nonverbal executive control.	Published	Sullivan, M. D., Janus, M., Moreno, S., Astheimer, L., & Bialystok, E. (2014). Early stage second-language learning improves executive control: evidence from ERP. *Brain and Language* 139, 84–98. [49]
Viswanathan, M., & Bialystok, E. (2007). Effects of bilingualism and aging in multitasking.	Not published	
Viswanathan, M. & Bialystok, E. (2007). Exploring the bilingual advantage in executive control: The role of expectancies.	Published	Viswanathan, M. (2015). Exploring the bilingual advantage in executive control: Using goal maintenance and expectancies. PhD thesis: York University. [50]

* A bachelor thesis by Simone Chin with the title “Working memory in bilinguals and second language learners” is listed in the records of the University of Central Florida, but the full text cannot be consulted as it is not publicly available online.

## Data Availability

No new data were created or analyzed in this study. Data sharing is not applicable to this article.

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
