# Peer review of "A Classification Bias and an Exclusion Bias Jointly Overinflated the Estimation of Publication Biases in Bilingualism Research"

_behavsci, 2023, doi:10.3390/bs13100812_

Round 1
Reviewer 1 Report
The paper presents new evidence regarding supposed publication biases in bilingualism research. It is truly well done, standing out as an example of clear and well structured scholarly writing. The author has masterfully organized their ideas, presenting a coherent and logical flow of information that is easy to follow and understand. Their meticulous attention to detail is evident in the way they seamlessly integrate theoretical issues, the methodological approach, and empirical findings into a cohesive narrative. This article undoubtedly holds great potential to make a significant impact on the field of bilingualism research.
As a reviewer, I am happy to report that I do not have any major comments or edits, as I feel like the paper is in a highly polished state. However, here are a couple minor notes I had:
- The discussion of the Jar Metaphor is well used and explained. However, the paragraph before it setting it up felt slightly disjointed organizationally. Perhaps just fleshing out the ideas presented here would help situate it better in the introduction that otherwise has a great flow.
- In the methods, the author uses the single category of "mixed results," whereas Bruin et al. (2015) originally had separated this into two categories based on whether they were predominantly positive or predominantly negative. Given how much of the paper focuses on the importance of not conflating negative and null results into one category, it is probably worthwhile to explain why this change to the categories was also undertaken.
- The inclusion of more varied types of published works is an important and strong argument of the paper. It is a bit surprising that it is only included in the discussion. To me it seems important enough to move as a second separate part of the main methods -- the first being the re-categorization and the second being the inclusion of these works.
Author Response
I thank the reviewer for their positive assessment. Please see the attachment for a point-by-point response to all the points raised by the reviewers.

Reviewer 2 Report
This is a most important article, which certainly contributes to a better understanding of alleged biases in scientific publishing. The reader is first led to see the problems with de Bruin et al.'s (2015) conclusions through a statistical analysis, and after that the reader learns about the authors' failure to consider the subsequent stages of the publication process of the articles they had looked at is an important observation. Of course, this delay/time aspect might be the topic of a future study; I think the article is good as such.
I also congratulate the author for choosing a highly informative title for the article: what the article is about is evident for the reader from the beginning.
Author Response

(The authors gave the same response as above.)

Reviewer 3 Report
I don't know when I have enjoyed interacting with a paper more than I have with this one. I think there are a number of issues that need attention, and I look forward to seeing how the author handles them.
First, it is not clear that the classification de Bruin et al used is materially problematic. Yes, they merged negative and null findings, but there are so few negative findings, and they split 50/50, so they cannot materially affect the outcome. Given the small number, I don't see that a Bayesian analysis makes any sense. It is presumably because there are so few negative results that de Bruin et al combined them with null results to begin with. I'd like to see some justification of the analysis the author performs.
The real demonstration that the classification isn't a problem is the fact that the author gets the same apparently null results when merging the two and updating the number of "published" results. The author says (p 16) "Τaking tables 1, 3 and 4 together, 33/40 abstracts in the category ‘positive results’ have been published by 2023, while for the category ‘negative & null results’, the published abstracts are 14/17, or 82.5% and 82.35% for each category respectively. Ιn other words, even if we maintain the original classification and the null/negative mega-category, the updated numbers do not replicate the result of de Bruin et al. (2015), regarding a significant effect of ‘category’ on publication status." Assuming that that is the case, there was no point to the author's original analysis separating negative and null results. This new analysis, with updated data, apparently says that there is no bias in what is published. I therefore don't see the point of the foregoing discussion. The new analysis renders the previous analysis unnecessary. There is a question about whether all of the materials that the author considers publications actually are publications. The author refers to the exclusion of 'grey' literature as improper, but provides no substantive discussion. One curious feature of the little discussion is that the author seems to be arguing that the published literature is biased compared to the unpublished ('grey') literature, which is what they were arguing against earlier in the paper. At two points the author appears to suggest that, indeed, null results are less likely to be published, and says that this is typical of scientific findings. It is typical, but I don't see the relevance. I don't think there is a claim that the bilingualism literature is special. Rather, I think the claim is that it is like other unsubstantiated findings.
Second, I am not sure why the author doesn't also run frequentist analyses. The original data presentation is significant via a chi-square test and other tests. What are the virtues of a Bayesian analysis such that we should prefer it to a frequentist analysis?
Third, it's a matter of interpretation whether the null results "fully challenge" the bilingual advantage hypothesis. The author used the mantra "absence of evidence is not evidence of absence" several times. (The mantra is supposedly due to Carl Sagan.) Like most categorical statements outside of math and physics, this does not bear a lot of scrutiny, despite its popularity in the bilingualism literature. For example, I think it's safe to say that elves, goblins, and fairies do not exist, even though we only have absence of evidence. Especially in the absence of a causal mechanism that lays out boundary conditions, repeated failure to find a phenomenon actually is some evidence that the phenomenon does not exist and that claims that it does are due to confounds. I am not claiming that there is no bilingualism effect. I am claiming that absence of evidence is some evidence of absence.
If only evidence going in the other direction – evidence favoring monolinguals – is going to count as negative evidence, then it is correct to say that there is no evidence. But most debunking of most claims does not occur because there is solid evidence in the other direction. Most debunking comes from a failure to replicate, and that is what we have with the bilingual literature. Contrary to the author's claim that "null results cannot challenge or disprove the alternative hypothesis", null results can and do challenge alternative hypotheses. They cannot disprove the hypothesis; proof only exists in mathematics and logic. Nothing disproves an empirical hypothesis.
Fourth, the speculation on p 12 that positive and null results on bilingualism are like a piece of wood that ranges from red to blue in color has no evidence behind it. It cannot revolutionize the field if there is no theory about that piece of wood and how it got the colors it got. Metaphors have to be substantiated with theory and data, and there is neither to buttress this metaphor. We have no idea what "trade-offs" there might be or how trade-offs would work, and the author acknowledges that.
Fifth, I do not find the jar metaphor relevant. A and B seem initially not to know about the other's data, given that each makes their claims on the basis of the particular data that they have. Then they are talking to each other and arguing. At that point, A and B would rationally conclude that the jar holds at least blue and red pencils, and that they probably need to keep taking pencils out until the jar is empty. If pencils are at issue, nothing prevents both of the colors from being present and there is no reason for either A or B to conclude that there is only one color present. As they keep drawing they find other colors, at which point they would rationally conclude that there are pencils of several different colors in the jar. The author mentions "seemingly contradictory results". The results are not contradictory at all. Blue and red do not contradict each other. Red and blue pencils could jointly make up the contents of the jar. Red and blue are not the equivalent of positive and null findings. No metaphor is completely apt, but I don't see any resemblance here to the bilingualism debate.
Sixth, the discussion of the funnel plot asymmetry suggests various reasons that the funnel plot might not warrant the conclusion de Bruin et al draw. But why not show that those reasons actually obtain, rather than speculate? Absent an actual demonstration, I don't see the value of simply pointing out the ways a funnel plot can be misleading. Is it misleading in the case of bilingualism or not?
In sum, I hope the author can address issues 1-6.
Author Response

(The authors gave the same response as above.)

Reviewer 4 Report
Summary:
The paper reassesses previous data and analyses of the impact of publication bias in the field of bilingualism research. The authors argue that methodological flaws in previous research have distorted the perception of the magnitude of publication bias effects.
By re-analysis of the data in Bruin et al. (2015) they identify a classification bias as the main culprit alongside with the observation that the former study neglected publication of some of the studies. In sum, such flaws may have led to exaggerated estimates of publication bias in bilingualism research.
In the main analysis of the current paper, the authors investigate what the publication bias would look like with a reclassification of studies considered in the paper by Bruin et al. (2015). Crucially, the authors argue to disentangle studies with null-results from such with negative outcome (i.e., studies providing evidence in favor for a bilingual disadvantage) – and not grouping them together as antagonists for positive outcomes (=the classification bias).
A Bayesian approach is used for the analysis.
The results show that the previous claim that publication bias has affected all studies that “fully challenge” the bilingual advantage hypothesis is not entirely true. The present study finds a publication bias only for studies that find evidence in favour of a bilingual advantage over studies that find null results. Crucially, studies that provide evidence against a bilingual advantage in the narrow sense (i.e., for a bilingual disadvantage) are not suspected of being published less frequently than studies with results in favour of bilingual advantages.
The authors further argue that a more precise examination and interpretation of different types of bias (such as classification bias) is a central key to a better understanding of the complex picture of bilingual advantages and disadvantages (and their interrelationships) - and thus to a better understanding of bilingualism itself.
Assessment:
The problem of publication bias is ubiquitous and well known in almost all (empirical) sciences.
It is therefore all the more gratifying to read a paper that takes a closer look at the problem and attempts to investigate the nature and impact of such bias(es).
Particularly in the field of bilingualism research (and language acquisition in general), the field can only benefit from such an analysis and subsequent self-reflection on its own blind spots.
I therefore believe that the present manuscript and its findings will be useful to most researchers in the field.
Formally, the paper is well written and follows scientific standards in writing, citation and presentation of statistical analysis. It also takes into account current and relevant literature.
Overall, I enjoyed reading the manuscript.
However, I have one concern that I feel should be addressed prior to publication. This concerns the organisation of the paper.
The authors focus their main analysis (method and results) on the recategorisation of studies previously reported in Bruin et al.
However, in the discussion section, the authors additionally introduce the problem that some studies that were labelled as "not published" by Bruin et al. have since been published (or were even published at the time). They also provide two alternative additional analyses with the updated dataset, maintaining the original classification etc.
I think that the right place for these aspects and this (in my opinion important) information should not be in the discussion section - but maybe in the main part as well together with the categorization problem (analysis, results) - maybe as a sub-section there.
Otherwise, if the authors are of the opinion that the central point of the paper is indeed the classification bias and the inclusion/exclusion of studies in the published group is only considered as an additional argument for a critical reflection of different biases in the discussion, I would suggest to reduce the prominence of the additional analyses in the discussion (and e.g. move details of the analyses, Figures 8 & 9, to an appendix?).
Minor / formal points:
Line 211: delete “provides”
Linees 234-235: you might add the BFs of both contrasts in the text also.
The size, resolution and position of some of the figures and tables in my copy of the manuscript do not seem to be optimal, e.g. the numbers in the figures are not always clearly legible.
Author Response

(The authors gave the same response as above.)

Round 2
Reviewer 3 Report
As before, I enjoyed engaging with this paper, and I appreciate the author's serious consideration of my objections. I will leave it up to the editor, of course, to decide whether the paper is good to go as it now stands.
I believe that there are some outstanding issues. Basically, I do not think that the situation with the nay-sayers is as dire as the author paints it, and I think it would help if there were some crisper conclusions. I concentrate here on four issues.
1) I agree that people do not generally run frequentist and Bayesian analyses together, but there is an issue when they give contradictory results. I think the author rather ignores that. But to me the main issue is, what should we make of the fact that, with null results in the initial analysis, the alternative hypothesis is rather likely?
2) The author thinks we should counterpose positive and negative findings. As I said in my first review, and as the author agrees, that is not generally what happens when a field decides that a supposed phenomenon is not an actual phenomenon. If we doubt stereotype threat, for example, it is not because people respond better rather than worse under stereotype threat, but because we do not see convincing examples of stereotype threat when confounds of various sorts are removed. The same argument is made in the bilingualism literature and I do not think that the author confronts it. Those who doubt that there is a cognitive benefit to bilingualism do so not because of the existence of cases where monolinguals are better than bilinguals (though at least a few such cases exist) but because when large samples are used, or when confounds are removed, there appears to be no phenomenon. I do not think that the author fully engages this issue.
3) I agree with the author that combining null and negative results is not a good move in general. My point was only that in this case it didn't seem to matter. I believe that the author agrees. I think it might be better to acknowledge that in deBruin et al's case it doesn't matter and also conclude that in general it's not a good practice.
I think classifying results as mixed/positive vs mixed/negative is not worth doing, in part because people tend not to report all of their results. I suspect that most studies that use more than one task have mixed results (at best). One sees many studies where data are not reported for some studies. Poor practice, and practice that I hope is on its way out, but certainly evident in many published papers.
Another issue in reported studies is what measures are used to decide whether there is an effect or not, and whether people are statistically correctly using conflict effect scores. E.g., for fast responders, a small difference between congruent and incongruent trials may be more meaningful than that small difference for slow responders. The literature has not grappled with this. And which congruent and incongruent scores should be used? There are often different combinations of blocks. Authors seldom report all the different analyses they performed. I am not suggesting that the author get into all of this, but I think it should lessen confidence in the reporting norms of the field.
4) I continue to see no justification for proposing that bilingualism results are like pencils that are blue at one end and red at another. The author doesn't try to apply the metaphor to any existing findings and I don't know of any to which the metaphor would apply. I think the metaphor posits something new for which there is no evidence at all. I don't know what would correspond to red and blue and everything in between. The author suggests that blue corresponds to positive results and red to negative results and that there is everything in between. Is the in between some positive and some negative? I cannot make sense of the metaphor. I do not see what that piece of wood corresponds to.
I don't understand why the author contrasts positive and negative studies. That is not generally what is at issue. What is at issue is positive vs null results and, as the author states elsewhere, that is generally what is at issue in studies where there is an initial flurry of positive results that is followed by null results. Null results are much more common than negative results, which is exactly what one would expect if there were no phenomenon.
In every bilingualism study there are many potential confounds, present to different degrees in each sample and in different samples at different times. There are also many different tasks, which by and large do not correlate with each other. I think the jar metaphor mischaracterizes the situation (to the extent that I can understand it).
Yes, there are plenty of reports of bilingualism effects on cognition. There are also plenty of reports of ESP, flying saucers, miracles, and various supernatural effects. People are free to believe all of those reports. But an absence of well-controlled large studies demonstrating any of them, combined with the absence of a mechanism, is some reason to doubt that they exist. When people doubt bilingualism effects on cognition, they do not think that monolinguals are superior. They think that it's like studies purporting to show positive effects of training short term memory, for which there is similar dispute. There are many confounding individual differences that can give rise to apparent effects.In one of the papers the author refers to, there is a reference to the vocabulary disadvantage of bilinguals compared to the cognitive advantage. Those are completely different phenomena. To my mind, it makes no sense to contrast them as if they are part of the same "stick".
The English is generally fine.
Author Response
I thank the reviewer for engaging with my work. Please see the attachment for a detailed point-by-point reply to the points they raised.
